# A Review of Evidence for the Involvement of the Circadian Clock Genes into Malignant Transformation of Thyroid Tissue

**Arcady A. Putilov** [1,2,*] **, Elena V. Budkevich** [1] **and Roman O. Budkevich** [1]

1 Laboratory of Nanobiotechnology and Biophysics, North-Caucasus Federal University,
355029 Stavropol, Russia; budkevich.ev@yandex.ru (E.V.B.); budkev@mail.ru (R.O.B.)

2 Laboratory of Sleep/Wake Neurobiology, Institute of Higher Nervous Activity and Neurophysiology of the
Russian Academy of Sciences, 117865 Moscow, Russia

* Correspondence: putilov@ngs.ru; Tel.: +7-49-30-53674643 or +7-49-30-61290031

**Abstract:** (1) Background: In 2013, the results of a pioneer study on abnormalities in the levels and circadian rhythmicity of expression of circadian clock genes in cancerous thyroid nodules was published. In the following years, new findings suggesting the involvement of circadian clock-work dysfunction into malignant transformation of thyroid tissue were gradually accumulating. This systematic review provides an update on existing evidence regarding the association of these genes with thyroid tumorigenesis. (2) Methods: Two bibliographic databases (Scopus and PubMed) were searched for articles from inception to 20 March 2023. The reference lists of previously published (nonsystematic) reviews were also hand-searched for additional relevant studies. (3) Results: Nine studies published between 2013 and 2022 were selected. In total, 9 of 12 tested genes were found to be either up- or downregulated. The list of such genes includes all families of core circadian clock genes that are the key components of three transcriptional–translational feedback loops of the circadian clock mechanism (*BMAL1*, *CLOCK*, *NPAS2*, *RORα*, *REV-ERBα*, *PERs*, *CRYs*, and *DECs*). (4) Conclusions: Examination of abnormalities in the levels and circadian rhythmicity of expression of circadian clock genes in thyroid tissue can help to reduce the rate of inadequate differential preoperative diagnosis for thyroid carcinoma.

**Keywords:** thyroid cancer; thyroid nodules; preoperative diagnostic; circadian clockwork dysfunction; circadian clock genes

## 1. Introduction

The first results pointing to the alternation of expression of circadian clock genes in human malignant thyrocytes were published 10 years ago [1]. In the following years, further evidence for the association of these genes with thyroid cell oncogenic transformation was gradually accumulating [2–9].

As stressed in several (nonsystematic) reviews of such findings [10–18], the examination of the pattern of expression of circadian clock genes might be of practical importance for the establishment of an adequate presurgical diagnosis of thyroid tumors [19]. Although the frequency of thyroid nodules (i.e., discrete masses in the gland) is very high (they are detected in up to 65% of general populations), most of them are benign. A relatively low percentage of cancerous nodules (about 5%) increases the risk of unnecessary surgeries in the case of asymptomatic benign nodules and increases the risk of delays with diagnosis and treatment in the case of asymptomatic cancerous nodules [20,21].

Thyroid cancers (TCs) are usually divided into five main histological types: "papillary," "follicular," "poorly differentiated," "undifferentiated/anaplastic," and "medullary" (PC, FC, PDTC, ATC, and MTC, respectively) [22,23]. The origin from neuroendocrine C cells distinguishes the last type (MTC) from other four types. Two of the remaining four types (PC and FC) are classified as well differentiated. An incomplete tumor capsule with

expansive growth is the most typical feature of PDTC, and among five histological types, ATC is regarded as the most aggressive form. The vast majority of TCs are classified as PC and FC types (85%), while only a small fraction of thyroid tumors (5–7%) lose features of cell origin and are classified as either PDTC or ATC [24]. The prognosis of patients with PC and FC is rather optimistic (e.g., they are usually curable with radioactive iodine therapy or surgery [25–27]), while the risk of mortality for ATC is high due to its rapid progression [28].

Until recently, fine-needle aspiration biopsy aimed at identification of thyroid malignancies has served as a safe and accurate tool for clinical evaluation of nonsecreting thyroid nodules [20]. Although surgery is not recommended without evaluating the results of such biopsies, the rate of unnecessary surgery remains high (e.g., after surgical interventions, 70–90% of thyroid cases were found to be benign [29]). The difficulty of distinguishing between benign and malignant thyroid nodules has been recognized as one of the most challenging issues of diagnostic approaches in oncology of the thyroid gland [30]. The rate of unnecessary surgeries and delays with diagnosis and treatment of thyroid cancer can be lowered by the development of preoperative markers for thyroid malignancies [31]. In order to improve preoperative diagnosis, the molecular testing platforms have been introduced. In efforts to reduce unnecessary surgical interventions, these platforms can be used as an integral part of the cytological evaluation in conjunction with fine-needle aspiration biopsy [32–36]. Therefore, it is critical to determine whether presurgical molecular biomarkers for thyroid carcinoma might include indices of abnormal functioning of circadian clocks [10–14].

The importance of the circadian clockwork for multicell organisms has become evident from findings indicating that almost each cell in almost each tissue type of almost each organism contains a set of genes involved in the construction and functioning of its own circadian clocks with a near 24 h (circadian) period [37,38].

A transcriptional–translational feedback loop is a key feature of this regulating mechanism [37–39]. In a mammalian cell, the circadian cycle is initiated by BMAL1 (brain and muscle aryl hydrocarbon receptor nuclear translocator-like), CLOCK (circadian locomotor output cycles kaput), and NPAS2 (neuronal PAS domain protein 2). Their proteins are members of the basic helix–loop–helix/Per-Arnt-Sim (bHLH-PAS) family of transcription factors. In the cytoplasm, they interact with each other to pro-duce heterodimers (BMAL1/CLOCK and BMAL1/NPAS2). At the next phase, these heterodimers translocate to the nucleus to activate the transcription of several other clock genes, including CRYs (cryptochrome circadian regulator 1 and cryptochrome circadian regulator 2), PERs (period circadian regulator 1, period circadian regulator 2, and period circadian regulator 3), and DECs (differentially expressed in chondrocyte 1 and differentially expressed in chondrocyte 2). BMAL1/CLOCK and BMAL1/NPAS2 heterodimers act as transcription factors and bind the E-box regions on the promoters of these target genes that in turn encode for the repressor components of the circadian clocks. At a later phase of the cycle, the protein products of these genes dimerize and form complexes between themselves (PERs with CRYs and DEC1 with DEC2). At the final phase, the feedback loop is completed by transporting these cytoplasmic dimers (PERs/CRYs and DEC1/DEC2) back into the nucleus to suppress activity of BMAL1/CLOCK and BMAL1/NPAS2 heterodimers. When they are suppressing activity of these heterodimers, they are also repressing their own expression, thus giving rise to a new cycle (i.e., the next cycle is started with allowing the transcription of BMAL1, CLOCK, and NPAS2) [38–40]. The circadian expression of BMAL1 and NPAS2 is also influenced by two nuclear receptors, REV-ERBα (or NR1D1, nuclear receptor subfamily 1 group D member 1) and RORα (RAR related orphan receptor alpha). Both receptors are activated by BMAL1/CLOCK, and they in turn regulate expression of BMAL1 and NPAS2 genes by acting on their promoters [41] (Figure 1).

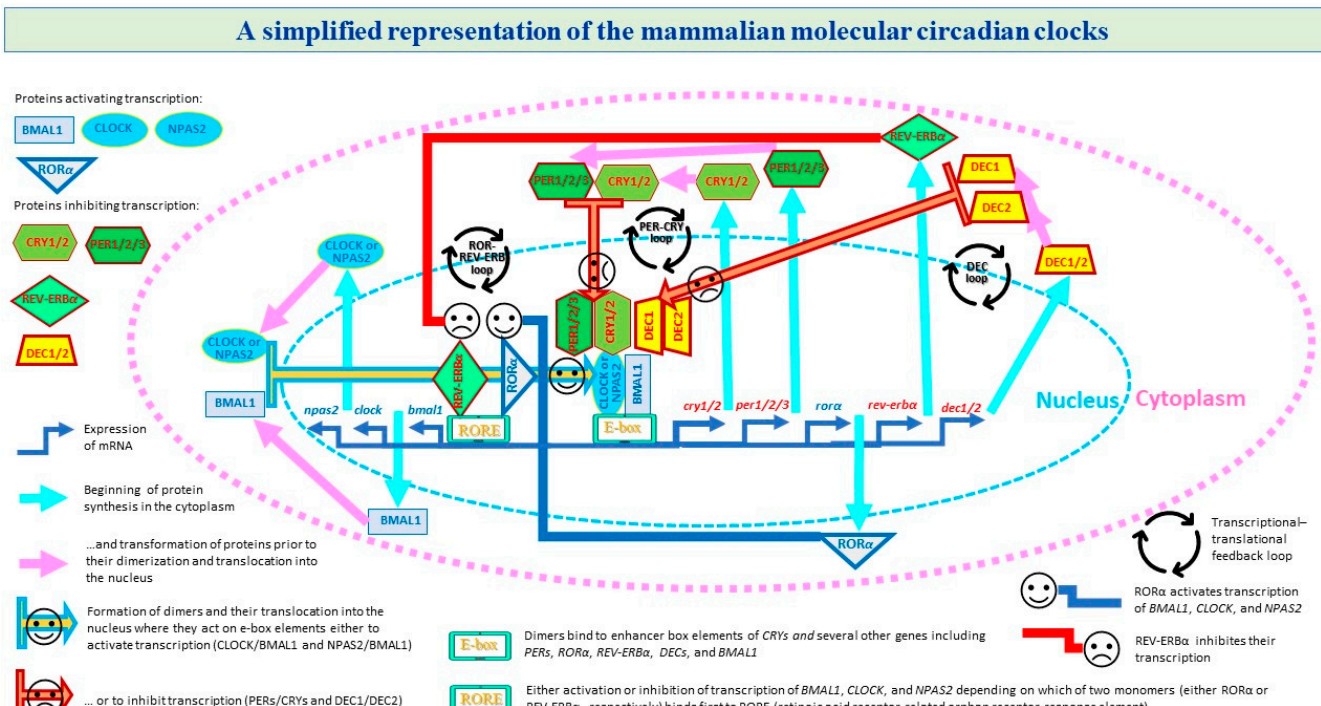

**Figure 1.** A simplified representation of three transcriptional–translational feedback loops. A transcriptional–translational feedback loop is a key feature of the circadian clock mechanism. In a mammalian cell, this mechanism includes several transcriptional–translational feedback loops involving clock genes. The circadian cycle is initiated by three circadian clock genes producing proteins BMAL1, CLOCK, and NPAS2 (brain and muscle aryl hydrocarbon receptor nuclear translocator-like 1, circadian locomotor output cycles kaput, and neuronal PAS domain protein 2, respectively) that are the members of the basic helix–loop–helix/Per-Arnt-Sim (bHLH-PAS) family of transcription factors. In the cytoplasm, they start to interact with each other to produce heterodimers BMAL1/CLOCK and BMAL1/NPAS2. At the next phase, these heterodimers translocate into the nucleus to activate the transcription of several other clock genes, including *CRYs*, *PERs*, (cryptochrome circadian regulator 1 and cryptochrome circadian regulator 2, period circadian regulator 1, period circadian regulator 2, and period circadian regulator 3), *DECs* (differentially expressed in chondrocyte 1 and differentially expressed in chondrocyte 2), and *BMAL1* itself. Namely, BMAL1/CLOCK and BMAL1/NPAS2 heterodimers act as transcription factors by binding the E-box regions on the promoters of the target genes. In turn, *CRYs*, *PERs*, *DECs* encode for the repressor components of the circadian clocks. At a later phase of the cycle, the protein products of these genes dimerize and form complexes between themselves (PERs with CRYs and DEC1 with DEC2). At the final phase, the core feedback loop (PER-CRY) is completed by transporting the cytoplasmic dimers PERs/CRYs back into the nucleus to suppress activity of BMAL1/CLOCK and BMAL1/NPAS2 heterodimers. When they are suppressing activity of these heterodimers, they are also repressing their own expression, thus giving rise to a new cycle (i.e., this next cycle is started with the initiation of transcription of *BMAL1*, *CLOCK*, and *NPAS2*). The cytoplasmic dimers DEC1 and DEC2 are included in another feedback loop (DEC). As with PER and CRY dimers, they translocate into the nucleus to suppress activity of BMAL1/CLOCK and BMAL1/NPAS2 heterodimers. One more transcriptional–translational feedback loop (ROR-REV-ERB) includes two nuclear receptors, REV-ERBα (nuclear receptor subfamily 1 group D member 1) and RORα (RAR-related orphan receptor alpha), that additionally contribute to the circadian expression of *BMAL1* and *NPAS2*. Both receptors are activated by BMAL1/CLOCK, and in turn regulate expression of *BMAL1* and *NPAS2* genes by acting on their promoter RORE (retinoic acid receptor-related orphan receptor response element) in the opposite direction, i.e., to either inhibit or activate transcription depending on which of two proteins binds first to the promoter (either REV-ERBα or RORα, respectively).

In addition to controlling each other's expression, these molecular regulators drive the rhythms of expression of many thousands of target genes. Circadian transcription factors interact with various corepressors, coactivators, and chromatin-associated factors, which in turn read or write or erase chromatin histone modification marks for activating or repressing transcription [42–45].

Overall, the cellular circadian clocks govern the expression of approximately half of the protein coding genes, thus providing coordination of circadian rhythms originating from different biochemical, physiological, and behavioral systems in the organism [46,47].

At any higher (multicell) hierarchical levels, the same set of circadian clock genes constitutes a basis of more complex circadian clocks with a similar molecular makeup. The list of such multicell clocks includes the peripheral clocks of various organs and tissue, the clocks of endocrine and physiological systems, and finally the central clocks in the supra-chiasmatic nuclei, which entrain the circadian rhythms within an organism and correct their phases by responding to environmental time cues [48,49].

Overall, the numerous modulating effects of the circadian clocks on gene expression, DNA repair, proliferation, cellular metabolism, inflammation, apoptosis, etc. [50] gave rise to hypotheses that the abnormal functioning of circadian clock genes may have various pathological consequences, including tumorigenesis [51–59]. In particular, it was assumed that the characteristics of clock genes signaling their abnormal functioning can serve as specific molecular markers of malignant transformation of thyroid tissue [10–18].

In 2018, Angelousi et al. [15] presented results of a systematic search for studies testing the association of circadian clock genes with different types of endocrine cancer. Results on thyroid cancer were reported in only one of 23 eligible publications [1]. A majority of new studies confirming this association were published more recently [5–9].

Therefore, the aim of the present systematic review was to summarize evidence supporting the assumption of abnormal expression of circadian clock genes in thyroid carcinoma.

## 2. Results

The main results of the literature search are summarized in Table 1. These indicated that each of the circadian clock genes mentioned in the Introduction was tested at least once in thyroid carcinoma between 2013 and 2022 (Table 1) and the results of such testing were reported in nine selected publications, in the pioneer publication of Mannic et al. (2013) [1] and eight following articles authored by Mond et al. (2014) [2], Chitikova et al. (2015) [3], Makhlouf et al. (2016) [4], Gallo et al. (2018) [5], Sadowski et al. (2019) [6], Lou et al. (2021) [7], Xu et al. (2022) [8], and Mou et al. (2022) [9] (article in Chinese with abstract in English). The authors of the nine selected publications interpreted the detected increase and decrease in expression of a gene compared to normal samples as its up- and downregulation, respectively. Most frequently, the transcripts of BMAL1 (brain and muscle aryl hydrocarbon receptor nuclear translocator-like) were studied under this and another name (ARNTL). In five published studies, the results suggested its upregulation. BMAL1 forms heterodimers with CLOCK (circadian locomotor output cycles kaput) and NPAS2 (neuronal PAS domain protein 2), whose transcripts were less frequently tested (in three studies and one study, respectively). The results reported in two studies and one study, respectively, suggested that they were (like BMAL) upregulated (Table 1). Upregulation was also reported for REV-ERBα (or NR1D1, nuclear receptor subfamily 1 group D member 1) and RORα (RAR-related orphan receptor alpha) in one of three and one of two studies, respectively (Table 1).

**Table 1.** Involvement of 12 circadian clock genes into thyroid carcinoma.

| Feedback Loop | Name *Gene* | Protein | Circadian Clock Transcript Upregulated | Downregulated | Nonsignificant | No Data |
|---|---|---|---|---|---|---|
| Monomers or dimers are promoters | | | | | | |
| All three | *BMAL1* or *ARNTL* | BMAL1 or ARNTL | FTC and PTC [1] PTC [3] PTC [6] TC [7] MTC [9] | | | [2,4,5,8] |
| All three | *CLOCK* | CLOCK | TC [7] MTC [9] | | PTC [3] | [1,2,4–6,8] |
| All three | *NPAS2* | NPAS2 | ATC [8] | | | [1–4,6,7,9] |
| ROR-REV-ERB | *RORα* | Nuclear receptor ROR-alpha | PTC [2] | | PTC [3] | [1,4–9] |
| Monomers or dimers are suppressors | | | | | | |
| ROR-REV-ERB | *REV-ERBα* or *NR1D1* | REV-ERBα | PTC [2] | | FTC, PTC [1] PTC [3] | [4–9] |
| PER-CRY | *CRY1* | CRY1 | | | FTC, PTC [1] PTC [3] TC [7] | [2,4–6,8,9] |
| PER-CRY | *CRY2* | CRY2 | | FTC and PTC [1] PDTC [4] TC [7] | PTC [3] | [2,5,6,8,9] |
| PER-CRY | *PER1* | PER1 | | | FTC, PTC [1] PTC [3] TC [7] | [2,4–6,8,9] |
| PER-CRY | *PER2* | PER2 | TC [7] | FTC and PDTC [4] | FTC, PTC [1] PTC [3] | [2,5,6,8,9] |
| PER-CRY | *PER3* | PER3 | | | FTC, PTC [1] PTC [3] | [2,4–9] |
| DEC | *DEC1* | DEC1 | | PTC [5] | | [1–4,6–9] |
| DEC | *DEC2* | DEC2 | | PTC [5] | | [1–4,6–9] |

Notes. TC: thyroid carcinoma. Malignant neoplasms: not further specified (TC), well-differentiated papillary (PTC), follicular (FTC), poorly differentiated (PDTC), undifferentiated/anaplastic (ATC). Thyroid C-cell–derived carcinoma: medullary (MTC). References (publication year): [1] Mannic et al. (2013); [2] Mond et al. (2014); [3] Chitikova et al. (2015); [4] Makhlouf et al. (2016); [5] Gallo et al. (2018); [6] Sadowski et al. (2019); [7] Lou et al. (2021); [8] Xu et al. (2022); [9] Mou et al. (2022). No data: this transcript of circadian clock genes was not tested/mentioned in this article. An increase and a decrease in expression compared to normal samples are interpreted as up- and downregulation, respectively. See Figure 1 for more details on the involvement of monomers (RORα and REV-ERBα) and dimers formed by BMAL1, CLOCK, NPAS2, PERs, CRYs, and DECs in the circadian clock mechanism.

Downregulation was consistently (in three of four articles) revealed for CRY2 (cryptochrome circadian regulator 2). DECs (differentially expressed in chondrocyte 1 and differentially expressed in chondrocyte 2) were tested in one study the results of which indicated that, similarly to the results on CRY2, these two genes were downregulated (Table 1). For only one gene, PER2 (period circadian regulator 2), the results on its transcripts were contradictory. This gene was reported to be upregulated in unspecified thyroid cancer (TC [7]) and downregulated in follicular and poorly differentiated cancer (FTC and PDTC [4]). The results on the transcripts of this gene were reported to be nonsignificant (Table 1) in two other studies of well-differentiated papillary cancer (PTC [1,3]) and follicular cancer (FTC [1]).

Nonsignificant results were obtained for the transcripts of three genes: CRY1 (cryptochrome circadian regulator 1) and PER1 (period circadian regulator 1) in three studies and for PER 3 (period circadian regulator 3) in two studies (Table 1).

The pioneer publication [1] contains results of additional comparisons of diurnal profiles of expression of seven circadian clock genes in thyrocytes cultured in vitro for 7 days and harvested every 6 h during 36 h (Table 2).

**Table 2.** Abnormal circadian rhythmicity of transcription of 7 clock genes in thyroid carcinoma.

| Name<br>*Gene* | Protein | Circadian Rhythmicity of Transcript | | Compared to<br>Other TC Type |
|---|---|---|---|---|
| | | Altered Phase | Altered Amplitude | |
| *BMAL1* or<br>*ARNTL* | BMAL1 or<br>ARNTL | PDTC | | PTC or HB |
| *REV-ERBα* or<br>*NR1D1* | REV-ERBα | PDTC | | PTC or HB |
| *CRY1* | CRY1 | PDTC | | PTC or HB |
| *CRY2* | CRY2 | PDTC | | PTC or HB |
| *PER1* | PER1 | | PDTC | PTC or HB |
| *PER2* | PER2 | PDTC | | PTC or HB |
| *PER3* | PER3 | PDTC | | PTC or HB |

Notes. Summary of the findings on diurnal profiles of expression of 7 circadian clock genes reported in Mannic et al. (2013) [1]. HB: healthy tissue or benign nodule. See other notes in Table 1.

The results showed that thyrocytes from healthy and benign (HB) thyroid nodules kept their circadian properties (i.e., the clock gene expressions in these nodules exhibited circadian oscillatory patterns in synchronized thyrocytes). In contrast, only alternated circadian profiles were detected in thyrocytes from poorly differentiated thyroid carcinomas (PDTCs) [1]. As shown in Table 2, Mannic et al. [1] revealed abnormal circadian phase positions for six of seven tested transcripts. Moreover, they failed to determine this position for the seventh transcript due to an abnormally low circadian amplitude [1]. Sadly, there have been no attempts so far to provide support to these very promising findings. Therefore, replicability and reliability of the abnormalities in circadian oscillatory patterns remain to be elaborated.

In sum, upregulation was reported for all three circadian clock genes involved in the circadian cycle in its earlier phases: BMAL1, CLOCK and NPAS2. These genes form heterodimers with the function of transcription activators. It was found that all three genes were upregulated in thyroid carcinoma [1,3,6–9] (Table 1). Upregulation was also shown for the genes producing receptors REV-ERBα and RORα, which are activated by BMAL1/CLOCK (Table 1). Downregulation was reported for CRYs, DECs, and PERs. These genes play the role of suppressors of activity of BMAL1, CLOCK and NPAS2 at later phases of the cycle. Downregulation of genes that form dimers with the function of transcription inhibitors was revealed for CRY2 [1,4,7], DEC1 and DEC2 [5], and PER2 ([4], but see contradicting results reported in [7]; Table 1).

Overall, either up- or downregulation was reported for 9 of 12 tested circadian clock genes. This result suggested that if two or more such genes were tested, there was a good chance of finding evidence of dysregulation of expression of at least one of them (Table 1).

## 3. Discussion

In 2013, Mannic et al. [1] published results of their pioneer study providing evidence for the involvement of circadian clockwork dysfunctions into malignant transformation of thyroid tissue [1]. Further evidence for the abnormal expression of circadian clock genes in cancerous thyroid nodules has gradually accumulated in the following 10 years [2–9]. The main aim of the present systematic review was to identify articles reporting results of testing the association between thyroid carcinoma and expression of circadian clock genes. The findings of nine relevant studies [1–9] were summarized and allowed the conclusion of plausibility of the assumption of abnormal expression of circadian clock genes in thyroid carcinoma. The reviewed findings provided evidence that if just two of these genes were tested, there was a 100% chance of revealing that expression of at least one of them was abnormal.

Additional results of the pioneer study of Mannic et al. (2013) [1] on the circadian patterns of gene expression indicated that these patterns were alternated in each of seven circadian clock genes tested in poorly differentiated thyroid nodules (PDTCs). Therefore,

the integration of such studies on circadian biology of the thyroid gland into management of thyroid cancer might lead to the development of new methodology for diagnosis of malignant transformation of thyroid tissue. It is plausible to expect that an examination of abnormalities in the levels and circadian rhythmicity of expression of circadian clock genes in this tissue can help to decrease the rate of inadequate differential preoperative diagnosis for thyroid carcinoma.

In more detail, upregulation of BMAL1 was reported for the first time in the pioneer study [1] and confirmed by results of four subsequent studies: Chitikova et al. (2015) [3], Sadowski et al. (2019) [6], Lou et al. (2021) [7] and Mou et al. (2022) [9]. Expression of this gene was not tested in the remaining four studies of Mond et al. (2014) [2], Makhlouf et al. (2016) [4], Gallo et al. (2018) [5], or Xu et al. (2022) [8], but the results of these studies allowed the extension of the list of circadian clock genes demonstrating abnormal expression in thyroid carcinoma. These results provided evidence for dysfunction of practically all core circadian clock genes in malignant thyroid tissue.

The only contradictory results were obtained for the transcription of one of 12 tested genes (PER2). Downregulation of PER2 was reported for follicular and poorly differentiated cancer (FTC and PDTC) [4], while upregulation was found in a study of unspecified thyroid cancer (TC) [7]. Results on other genes were not contradictory. Overall, upregulation was found for BMAL1, CLOCK, NPAS2, REV-ERB$\alpha$, and ROR$\alpha$, and downregulation was revealed for CRY2, DEC1, and DEC2.

Such results regarding the directions of changes in gene expression in thyroid carcinoma (increased or decreased transcription) were in good agreement with the directions expected from the roles of the transcripts of the circadian clock genes in the transcriptional–translational feedback loop regulating the body clocks (see Introduction). The general rule was that if the dimers of genes acted as promoters of transcription of other circadian clock genes, they were found to be upregulated, while if the dimers of genes suppressed transcription of other circadian clock genes, they were found to be downregulated.

In more detail, dimers with opposite functions of transcription activator and transcription inhibitor are formed by BMAL1, CLOCK and NPAS2 and by CRYs, DECs, and PERs, respectively. Therefore, if it were consistently found in the reviewed studies that BMAL1, CLOCK and NPAS2 were upregulated in thyroid carcinoma [1,3,6–9], the suppressors of activity of BMAL1, CLOCK and NPAS2 (CRYs, DECs, and PERs) would be expected to be downregulated in this carcinoma. Indeed, this expectation was confirmed by the reports indicating downregulation of the majority of genes controlling expression of BMAL1, CLOCK and NPAS2. Upregulation was revealed for CRY2 [1,4,7], DEC1 and DEC2 [5], and PER2 ([4] but not [7]; see above).

Moreover, upregulation was obtained for the genes coding REV-ERB$\alpha$ and ROR$\alpha$ monomers [2], which are activated by BMAL1 and CLOCK heterodimer.

In oncological diseases, patient prognosis might be potentially improved by normalization of circadian rhythms [18]. Therefore, chronobiotic therapeutic factors were recommended to be used for prevention and/or treatment of these diseases (e.g., [17]). Two major treatment approaches were suggested for avoiding the pathological consequences of circadian clock dysfunction. The first is a direct pharmacological activation of some circadian clock genes [60]. Another is a restoration of normal circadian rhythms with the help of small molecules that are able to modulate activity of the core components of circadian clocks [61]. However, such studies remain at a stage of basic experimental exploration.

It is well established that thyroid-stimulating hormone (TSH) produced by the pituitary gland plays a pivotal role in regulating the hypothalamic–pituitary–thyroid axis. Therefore, this hormone serves as the major marker of hormonal and physiological activity of the pituitary and thyroid gland [13]. It was proposed that elevated TSH levels in serum can have diagnostic value in the presurgical management of thyroid carcinoma [62–68], and that the measurement of these levels can be recommended as an easily performed additional tool for decision-making in patients with indeterminate cytological findings [69–71]. Moreover, TSH exhibits a robust 24 h rhythm of secretion, and the diurnal profile of this hormone

in serum is regarded as one of three most reliable markers of the circadian rhythmicity of the human organism [10,72–75]. TSH is rhythmically secreted in response to the neuronal and humoral signals from the central clocks, but thyroid tissue that contains the same set of circadian clock proteins as the set expressed in the central clocks also exerts its influence on regulation of this hormone [76,77]. Therefore, it is plausible to suggest that thyroid cancer is linked to both an abnormal circadian rhythm of TSH and an abnormal expression of the circadian clock genes. Empirical support of this suggestion might lead to the recommendation of evaluation of both this hormone rhythmicity and these genes' expression. However, with the exception of one publication [78], the literature is lacking studies examining the 24 h rhythm of TSH in patients with different types of thyroid carcinoma [12]. Therefore, it remains unknown whether the levels and daily pattern of TSH in serum or thyroid issue might be challenged in parallel with the levels and daily pattern of expression of circadian clock genes.

TSH directly bound to its receptor (TSH-R) [79] and levels of TSH circulating in thyroid tissue regulate the receptor signaling in the thyrocyte's membrane [80]. Studies suggested the involvement of TSH-R in thyroid cancer (e.g., [81–84]). In particular, a reduced number of binding sites was found in carcinoma tissue [85], and levels of TSH-R gene expression were shown to be lower in such tissue [82,86]. Expression was decreased and even lost in poorly differentiated and undifferentiated thyroid cancer, respectively [87–91]. Therefore, it was proposed that TSH-R might have important diagnostic value for thyroid cancer, e.g., as a marker of thyroid differentiation [84–93]. However, almost nothing has been published about the diurnal profile of TSH-R expression, and this expression has not been examined in relation to expression of any circadian clock genes [12].

Therefore, future research might be aimed at testing whether there are associations between pathological changes in levels and diurnal profiles of thyroid hormones, their receptors, and transcripts of circadian clock genes. For instance, further in vitro experiments on cultured malignant thyrocytes might examine whether thyroid tumorigenesis is linked to parallel pathological changes in regulation of circadian clock genes and circadian rhythmicity of TSH and its receptor (TSH-R) [12].

Environmental factors can contribute to the risks of both thyroid cancer and circadian dysregulation. Several types of endocrine cancers were found to be associated with chronic circadian disruption caused by travel across time zones, shift and night work, sleep insufficiency, and irregular sleep–wake cycles [94]. Patients with more severely disturbed sleep and/or circadian disruption have worse prognosis than those having good sleep and normal circadian rhythmicity [15]. In respect to thyroid cancer, night and shift work was reported to be associated with an increased risk of thyroid nodules [95]. Moreover, disturbances of the sleep–wake cycle were shown to be linked to a higher risk of thyroid cancer in postmenopausal nonobese women [96]. Such a higher risk was also reported for women who used a sedative–hypnotic drug [97].

Light at night was found to be associated with an increased risk of thyroid cancer [98]. This association may be partially driven by melatonin's deficiency leading to a decrease in its tumor suppression function and to the disruptions of sleep and circadian rhythms [98]. Future studies may provide deeper insights into the potential roles of melatonin in thyroid cancer etiology and into the biological pathways underlying the relationship between light at night and thyroid cancer.

Several traits of the sleep–wake cycle, such as sleep duration, insomnia, and chronotype, may be risk factors for cancer [99]. Lou et al. [7] found that patients with thyroid cancer reported sleep disturbances more often when they also expressed elevated levels of CLOCK, BMAL1, and PER2 and reduced levels of CRY2 compared to age-matched cancer-free controls. Evidence for the extensive cross-talk between sleep, circadian rhythms, and metabolic pathways involved in malignancy suggests a possibility for recommending screening cancer patients for sleep and circadian disruptions [100,101].

Given that several variants of circadian clock genes were found to be associated with incidents of thyroid carcinoma (e.g., [102–104]), polymorphism of human circadian clock

genes might also contribute to the risks of development of thyroid cancer. These associations and phenotypic traits underpinned by these variants require further exploration.

The mechanisms underlying the links between abnormalities of the circadian clock machinery and malignant transformation of thyroid tissue remain to be elucidated. Some proposed causes of the associations between circadian disturbances and cancer (including thyroid cancer) were discussed in several previously published reviews (e.g., [10,11,16,18,51–58]). Analysis of existing evidence [105] indicated that the circadian clocks may be dysregulated in many forms of cancer, but such dysregulation is not caused solely by inactivation of core circadian clock genes. Rather, it is accompanied by large-scale changes in circadian gene expression and coexpression because the cellular circadian clocks play an important role in tumorigenesis, tumor immune escape, tumor growth, metastasis, and the processes of regulation of proliferation, apoptosis, intracellular metabolism, etc.

As mentioned in the Introduction, the risks of unnecessary surgeries and delays to diagnosis and treatment of thyroid carcinoma remain high due to a relatively low percentage of cancerous nodules [20,21]. This problem was addressed in recent years by introducing two new approaches to the diagnosis of cancerous nodules: liquid biopsy and molecular testing platforms. Liquid biopsy provided a possibility to eliminate the invasive procedures needed to obtain tissue samples. Because such biopsy detects and analyzes biological samples released from the tumor into the bloodstream, it can be repeatedly performed in a noninvasive way, at lower cost and without the risks associated with classic tissue biopsy [30,106]. Using molecular platforms for stratification of patients with neoplasm provides more definitive guidance for decision-making in the clinic, including the decision to avoid unnecessary surgical interventions [32–36]. As already emphasized in previously published reviews [10–18], the examination of expression of circadian clock genes in thyroid carcinoma might be of practical importance for adequate presurgical diagnosis of thyroid cancer. The reviewed findings here allow a recommendation to be made for trying to improve the preoperational management of thyroid carcer by examining whether the levels and circadian rhythmicity of expression of these genes are altered in thyroid nodules.

This review had several limitations. The number of studies selected for this review was small. Therefore, we cannot evaluate replicability of these studies. Although all five main histological types of thyroid carcer (PC, FC, PDTC, ATC, and MTC) were tested in nine reviewed studies, their results remain insufficient for comparisons of these types to the extent of abnormal changes in expression of circadian clock genes. Some of these types (e.g., PDTC and ATC) might be more vulnerable than other types to the development of circadian clockwork disfunction. Therefore, the results obtained on one type should be generalized to other types with caution. We reviewed here only studies on expression of core circadian genes in thyroid cancerous nodules. Consequently, we cannot compare the results on down- and upregulation of these genes in these nodules with the findings on down- and upregulation of the same genes in other endocrine cancers. Moreover, this review was limited to studies on expression of core circadian clock genes. We did not search for studies exploring the relationships of these genes with genes playing an important role in the processes of regulation of proliferation, apoptosis, and intracellular metabolism of thyroid tissue. These relationships might be addressed in future reviews.

## 4. Materials and Methods

The main aim of this systematic review was to identify publications reporting results of testing significance of associations between thyroid carcinoma and the levels and diurnal pattern of expression of core circadian clock genes. PRISMA (Preferred Reporting Items for Systematic Reviews and Meta-Analyses) guidelines for systematic review were followed [107]. Two bibliographic databases (Scopus and PubMed) were searched for articles from inception to 20 March 2023. The reference lists of previously published nonsystematic reviews [10–14,16–18] were also hand-searched for additional relevant articles.

Syntax for conducting the search within Scopus was the following:

TITLE-ABS-KEY (((circadian AND transcription) OR (clock AND expression) OR (clock AND genes) OR (circadian AND genes)) AND ((thyroid AND cancer) OR (thyroid AND tumor) OR (thyroid AND nodule) OR (thyroid AND neoplasm)))

For PubMed, the same search syntax was applied:

((circadian transcription) OR (clock expression) OR (clock genes) OR (circadian clock)) AND ((thyroid cancer) OR (thyroid nodule) OR (thyroid tumor) OR (thyroid neoplasm))

In total, 118 articles were identified (the study selection process is reported in Figure 2 in the PRISMA format). The hand search in the reference lists of previously published reviews [10–18] gave two additional relevant studies [2,5] that were previously mentioned in some such reviews [10–12].

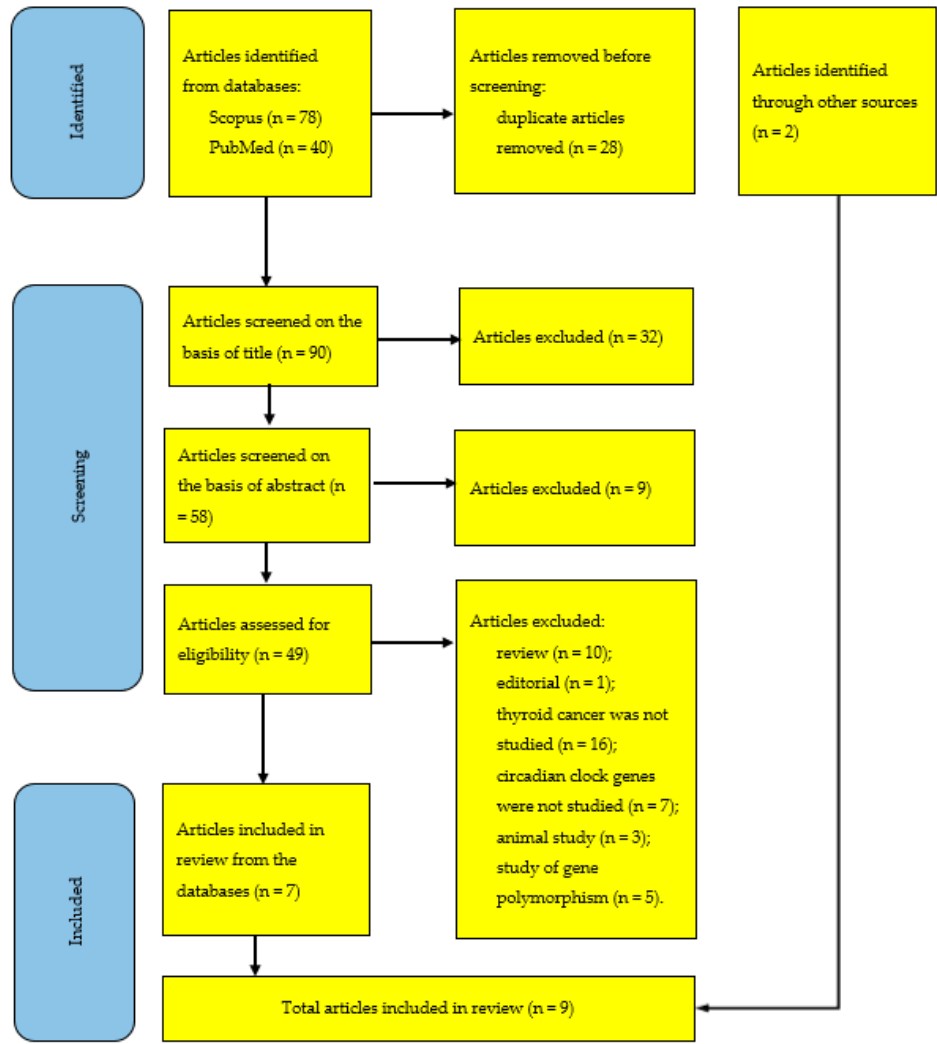

**Figure 2.** PRISMA flow diagram for study selection.

After exclusion of 28 duplicates, 23 and 9 articles were additionally excluded on the basis of their title and abstract, respectively. The assessment for eligibility was performed by applying the following exclusion criteria: (1) review ($n = 10$), (2) editorial ($n = 1$), (3) thyroid cancer was not studied ($n = 16$), (4) circadian clock genes were not studied ($n = 7$), (5) animal study ($n = 3$), and 6) study of gene polymorphism ($n = 5$).

Seven and two potentially eligible full-text articles [1–9] were finally selected by searching the databases and reference lists, respectively (Figure 2).

## 5. Conclusions

The first result suggesting the involvement of circadian clockwork dysfunction in thyroid tumorigenesis was published in 2013. In the following 10 years, further evidence for abnormalities in expression of circadian clock genes in thyroid carcinoma has been gradually accumulating. The present systematic review of existing evidence for such abnormalities found nine studies published from 2013 to 2022. In each of the studies, an abnormal expression in malignant thyroid tissue was documented for one or more circadian clock genes. In total, 9 of 12 tested genes were found to be either up- or downregulated. The list of up- or downregulated genes includes all families of core circadian clock genes, *BMAL1*, *CLOCK*, *NPAS2*, *CRYs*, *PERs*, *DECs*, *REV-ERBα*, and *RORα*, constituting all three transcriptional–translational feedback loops of the circadian clock mechanism. These findings allowed the conclusion that the assumption of association of thyroid carcinoma with circadian clockwork dysfunctions was supported. Therefore, the integration of circadian biology into management of thyroid cancer can improve the methods of preoperative diagnosis of thyroid cancer. In particular, such diagnosis might account for the results of evaluation of circadian rhythmicity and mean levels of expression of circadian clock genes.

**Author Contributions:** A.A.P.: conceptualization, formal analysis, investigation, methodology, project administration, supervision, validation, visualization, writing—original draft, review and editing. E.V.B.: investigation, writing—review and editing. R.O.B.: investigation, funding acquisition, writing—review and editing. All authors have read and agreed to the published version of the manuscript.

**Funding:** This research received no external funding.

**Institutional Review Board Statement:** Not applicable.

**Informed Consent Statement:** Not applicable.

**Data Availability Statement:** Not applicable.

**Acknowledgments:** The North-Caucasus Federal University also provided technical and other similar support to R.O.B., E.V.B. and A.A.P.

**Conflicts of Interest:** The authors declare no conflict of interest.

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
