# Peer review of "A Review of Evidence for the Involvement of the Circadian Clock Genes into Malignant Transformation of Thyroid Tissue"

_2624-5175, doi:10.3390/clockssleep5030029_

Round 1

Reviewer 1 Report

This systematic review aims to provide an update on existing evidence regarding the association of circadian clock dysfunction and circadian clock genes' expression alterations with thyroid tumorigenesis. It concludes that evaluation of circadian rhythmicity and mean levels of expression of circadian clock genes might have potential to improve preoperative diagnosis of thyroid cancer.

The review is focused on a relevant topic. The manuscript is written clear, presented in a well-structured manner and comprehensive. The aim is clearly defined, the introduction is sufficient and methodology adequate. The results are well presented and the conclusions are consistent with the evidence presented. 

Just minor:

Page 7, line 266-267: "It is well-established that thyroid-stimulating hormone (TSH) produced by the thyroid gland plays a pivotal role in regulating the hypothalamic–pituitary–thyroid axis" - This sentence should be rephrased since TSH is produced by pituitary gland not the thyroid.

Author Response

Reply to Reviewer #1Начало формыНачало формы

Comments and Suggestions for Authors

This systematic review aims to provide an update on existing evidence regarding the association of circadian clock dysfunction and circadian clock genes' expression alterations with thyroid tumorigenesis. It concludes that evaluation of circadian rhythmicity and mean levels of expression of circadian clock genes might have potential to improve preoperative diagnosis of thyroid cancer.

The review is focused on a relevant topic. The manuscript is written clear, presented in a well-structured manner and comprehensive. The aim is clearly defined, the introduction is sufficient and methodology adequate. The results are well presented and the conclusions are consistent with the evidence presented. 

Just minor:

Page 7, line 266-267: "It is well-established that thyroid-stimulating hormone (TSH) produced by the thyroid gland plays a pivotal role in regulating the hypothalamic–pituitary–thyroid axis" - This sentence should be rephrased since TSH is produced by pituitary gland not the thyroid.

Reply. Oh, yes, thank you very much, indeed, for noticing this. As recommended, the sentence was corrected.

Reviewer 2 Report

Association of the circadian clockwork dysfunctions and malignant transformation of thyroid tissue is the main target of current review article. Please conduct the concerns below.

1.      Diagnosis of benign and cancerous nodules (about 5%) in thyroid nodules is one of the most challenging issues in the search of articles related to current report. How to solve this problem? Please introduce in in detail.

2.      Abnormal circadian rhythmicity of the transcription of 7 clock genes in thyroid carcinoma indicated in Table 2 from one reference by Mannic et al. (2013) only. Is it reliable? Please describe in detail.

3.      Role of the thyroid-stimulating hormone (TSH) is important after activation of receptor (TSH-R) in the regulation of circadian rhythm and expression of the circadian clock genes. However, the reports seem not enough to support the hypothesis. How to solve this problem in advance?

4.      Melatonin is involved in sleep disturbances and/or circadian clockwork. But it was not conducted in current report. Why?

5.      In conclusion, 9 of 12 tested genes were found. But the mentioned 9 genes were not described in detail.

6.      Limitation(s) of current report will be helpful.

Author Response

Reply to Reviewer #2

Comments and Suggestions for Authors

Association of the circadian clockwork dysfunctions and malignant transformation of thyroid tissue is the main target of current review article. Please conduct the concerns below.

  1. Diagnosis of benign and cancerous nodules (about 5%) in thyroid nodules is one of the most challenging issues in the search of articles related to current report. How to solve this problem? Please introduce in in detail.

Reply #1. The details on how to solve this problem were provided in a special paragraph of Discussion:

 As it was mentioned in Introduction, the risks of unnecessary surgeries and delays with diagnosis and treatment of thyroid carcinoma remain high due to a relatively low percentage of cancerous nodules [20,21]. This problem was addressed in the recent years by introducing two new approaches to the diagnostic of cancerous nodules, liquid biopsy and molecular testing platforms. Liquid biopsy provided a possibility to eliminate the invasive procedures needed to obtain tissue samples. Because such biopsy detects and analyzes biological samples released from the tumor into the bloodstream, it can be repeatedly performed in a noninvasive way, at lower cost and without the risks associated with classic tissue biopsy [30,106]. Using the molecular platforms for stratification of patients with neoplasm provides more definitive guidance for the decision making in clinic including the decision to avoid unnecessary surgical interventions [32–36]. As it has been already emphasized in the previously published reviews [10-18], the examination of expression of circadian clock genes in thyroid carcinoma might be of practical importance for adequate presurgical diagnostic of thyroid cancer. The reviewed here findings allow the recommendation to try to improve the preoperational management of thyroid carcer by examining whether the levels and circadian rhythmicity of expression of these genes are altered in thyroid nodules.

  1. Pogliaghi, G. Liquid biopsy in thyroid cancer: From circulating biomarkers to a new prospective of tumor monitoring and therapy. Minerva Endocrinol. 2021, 46, 45–61. https://doi.org/10.23736/S2724-6507.20.03339-8.
  2. Abnormal circadian rhythmicity of the transcription of 7 clock genes in thyroid carcinoma indicated in Table 2 from one reference by Mannic et al. (2013) only. Is it reliable? Please describe in detail.

Reply #2. The details on how to solve this problem were provided by the extending this paragraph, and a sentence concerning reliability was also added.

…As shown in Table 2, Mannic et al. [1] revealed the abnormal circadian phase positions for 6 of 7 tested transcripts, and, moreover, they failed to determine this position for the 7th transcript due to an abnormally low circadian amplitude [1]. Sadly, there have been no any attempts, so far, to provide support to these very promising findings. Therefore, replicability and reliability of the abnormalities in circadian oscillatory patterns remains to be elaborated.

  1. Role of the thyroid-stimulating hormone (TSH) is important after activation of receptor (TSH-R) in the regulation of circadian rhythm and expression of the circadian clock genes. However, the reports seem not enough to support the hypothesis. How to solve this problem in advance?

Reply #3. Our suggestion about the possible way of solving this problem was given in a separate paragraph of Discussion

Therefore, future research might be aimed on testing whether there are associations between pathological changes in levels and diurnal profiles of thyroid hormones, their receptors, and transcripts of circadian clock genes. For instance, further in vitro experiments on cultured malignant thyrocytes might examine whether thyroid tumorigenesis might be linked to the parallel pathological changes in regulation of circadian clock genes and circadian rhythmicity of TSH and its receptor (TSH-R) [12].

  1. Melatonin is involved in sleep disturbances and/or circadian clockwork. But it was not conducted in current report. Why?

Reply #4. When discussed the environmental influences on the risks for thyroid cancer, we included in the revised version of the manuscript a special paragraph about hypothetical involvement of melatonin as both tumor suppression and transductor of light signal in the circadian and sleep networks.

Light at night was found to be associated with an increased risk of thyroid cancer [98]. This association may be partially driven by melatonin's deficiency leading to the decrease of its tumor suppression function and to the disruptions of sleep and circadian rhythms [98]. Future studies may provide deeper insights into the potential roles of melatonin in thyroid cancer etiology and into the biological pathways underlying the relationship between light at night and thyroid cancer.  

  1. Zhang, D.; Jones, R.R.; James, P.; Kitahara, C.M.; Xiao, Q. Associations between artificial light at night and risk for thyroid cancer: A large US cohort study. Cancer. 2021, 127:1448-1458. doi: 10.1002/cncr.33392.
  2. In conclusion, 9 of 12 tested genes were found. But the mentioned 9 genes were not described in detail.

Reply #5. We did, in many sections, thank you for this and other suggestions. Namely, to describe these genes in more details, we 1) enlarged this part of Abstract, 2) enlarged this part of Conclusions, 3) enlarged Table 1 and the notes to this table, and, finally, 4) added one more figure (Figure 1) with the details (both in the figure legend and as the icons of mentioned genes and their products in Figure 1A-1D).

1)

Abstract: …The list of such genes includes all families of core circadian clock genes that are the key components of three transcriptional-translational feedback loops of the circadian clock mechanism (BMAL1, CLOCK, NPAS2, RORα, REV-ERBα, PERs, CRYs, and DECs).

2)

  1. Conclusions

... The list of up- or downregulated genes includes all families of core circadian clock genes, BMAL1, CLOCK, NPAS2, CRYs, PERs, DECs, REV-ERBα, and RORα, constituting all three transcriptional-translational feedback loops of the circadian clock mechanism.

3) (notes to Table 1)

Notes. …See Figure 1 for more details on the involvement of monomers (RORα and REV-ERBα) and dimers formed by BMAL1, CLOCK, NPAS2, PERs, CRYs, and DECs in the circadian clock mechanism.

4) (Figure 1 legend)

Figure 1. A simplified representation of three transcriptional-translational feedback loops.

A transcriptional-translational feedback loop is a key feature of the circadian clock mechanism. In a mammalian cell, this mechanism includes several transcriptional–translational feedback loops involving clock genes. The circadian cycle is initiated by three circadian clock genes producing proteins BMAL1, CLOCK, and NPAS2 (brain and muscle aryl hydrocarbon receptor nuclear translocator like 1, circadian locomotor output cycles kaput, and neuronal PAS domain protein 2, respectively) that are the members of the basic helix-loop-helix/Per-Arnt-Sim (bHLH-PAS) family of transcription factors. In the cytoplasm, they start to interact with each other to produce heterodimers BMAL1/CLOCK and BMAL1/NPAS2. At the next phase, these heterodimers translocate into the nucleus to activate the transcription of several other clock genes including CRYs, PERs, (cryptochrome circadian regulator 1 and cryptochrome circadian regulator 2, period circadian regulator 1, period circadian regulator 2, and period circadian regulator 3), DECs (differentially expressed in chondrocyte 1 and differentially expressed in chondrocyte 2), and BMAL1 itself. Namely, BMAL1/CLOCK and BMAL1/NPAS2 heterodimers act as transcription factors by binding the E-box regions on the promoters of the target genes. In turn, CRYs, PERs, DECs encode for the repressor components of the circadian clocks. At a later phase of the cycle, the protein products of these genes dimerize and form complexes between themselves (PERs with CRYs and DEC1 with DEC2). At the final phase, the core feedback loop (PER-CRY) is completed by transporting the cytoplasmic dimers PERs/CRYs back into the nucleus to suppress activity of BMAL1/CLOCK and BMAL1/NPAS2 heterodimers. When they are suppressing activity of these heterodimers, they are also repressing their own expression thus giving rise to a new cycle (i.e., this next cycle is started with the initiation of transcription of BMAL1, CLOCK, and NPAS2). The cytoplasmic dimers DEC1/DEC2 are included in another feedback loop (DEC). Like PERs and CRYs dimers, they translocate into the nucleus to suppress activity of BMAL1/CLOCK and BMAL1/NPAS2 heterodimers. One more transcriptional–translational feedback loop (ROR-REV-ERB) includes two nuclear receptors, REV-ERBα (nuclear receptor subfamily 1 group D member 1) and RORα (RAR related orphan receptor alpha), that additionally contribute to the circadian expression of BMAL1 and NPAS2. Both receptors are activated by BMAL1/CLOCK, and, in turn, regulate expression of BMAL1 and NPAS2 genes by acting on their promoter RORE (retinoic acid receptor-related orphan receptor response element) in the opposite directions, i.e., to either inhibit or activate transcription depending on which of two proteins binds first to the promotor (either REV-ERBα or RORα, respectively).

  1. Limitation(s) of current report will be helpful.

Reply #6. The following limitation paragraph was added as the final part of Discussion:

This review had several limitations. The number of studies selected for this review was small. Therefore, we cannot evaluate replicability of these studies. Although all five main histological types of thyroid carcer (PC, FC, PDTC, ATC, and MTC) were tested in 9 reviewed studies, their results remain insufficient for the performing comparisons of these types on the extend of abnormal changes in expression of circadian clock genes. Some of these types (e.g., PDTC and ATC) might be more vulnerable than other types to the development of circadian clockwork disfunction. Therefore, the results obtained on one type should be generalized into other types with caution. We reviewed here only studies on expression of core circadian genes in the thyroid cancerous nodules. Consequently, we cannot compare the results on down- and upregulation of these genes in these nodules with the finding on down- and upregulation of the same genes in other endocrine cancers. Moreover, this review was limited to the studies on expression of core circadian clock genes. We did not search for studies exploring the relationships of these genes with genes playing an important role in the processes of regulation of proliferation, apoptosis, and intracellular metabolism of thyroid tissues. These relationships might be addressed in future reviews.
